# Video Dynamics Prior: An Internal Learning Approach for Robust Video Enhancements

**Gaurav Shrivastava**
University of Maryland, College Park
gauravsh@umd.edu

**Ser-Nam Lim**
University of Central Florida
sernam@ucf.edu

**Abhinav Shrivastava**
University of Maryland, College Park
abhinav@cs.umd.edu

## Abstract

In this paper, we present a novel robust framework for low-level vision tasks, including denoising, object removal, frame interpolation, and super-resolution, that does not require any external training data corpus. Our proposed approach directly learns the weights of neural modules by optimizing over the corrupted test sequence, leveraging the spatio-temporal coherence and internal statistics of videos. Furthermore, we introduce a novel spatial pyramid loss that leverages the property of spatio-temporal patch recurrence in a video across the different scales of the video. This loss enhances robustness to unstructured noise in both the spatial and temporal domains. This further results in our framework being highly robust to degradation in input frames and yields state-of-the-art results on downstream tasks such as denoising, object removal, and frame interpolation. To validate the effectiveness of our approach, we conduct qualitative and quantitative evaluations on standard video datasets such as DAVIS, UCF-101, and VIMEO90K-T.[1]

## 1 Introduction

Recently, video enhancement [54, 42, 43, 1, 12, 18, 49, 32] and editing [56, 11, 24, 50] tasks have attracted much attention in the computer vision community. Most of these tasks are ill-posed problems, i.e., they do not have a unique solution. For example, enhancement tasks such as video frame interpolation can have infinitely many plausible solutions. Therefore, the aim is to find a visually realistic solution coherent in both the space and time domain for such tasks.

Priors play a critical role in finding reasonable solutions to these ill-posed problems by learning spatio-temporal constraints. In recent years, deep learning has emerged as the most promising technique [54, 13, 39, 38, 4, 2, 43, 49, 22] for leveraging data to create priors that enforce these spatio-temporal constraints. However, the generalization of these data-driven priors mainly relies on data distribution at training time.

In the current era of short video platforms such as TikTok, Instagram reels, YouTube shorts, and Snap, millions of new artistic elements (e.g., new filters, trends, etc.) are introduced daily. This trend causes substantial variations between the train and the test sets, which results in diminished effectiveness of the learned prior. Moreover, frequent fine-tuning or frequent training to improve the efficacy of the prior significantly hamper the scalability of such models.

---

[1]Navigate to the webpage for video results.

37th Conference on Neural Information Processing Systems (NeurIPS 2023).

Furthermore, the existing video enhancement approaches assume clean and sharp input frames, which may not always be the case due to factors like bad lighting conditions during the scene capture or video compression for internet transmission. These factors introduce noise in the clean video signal, and feeding such noisy input frames to video enhancement approaches results in suboptimal enhancements.

To address these challenges, we propose a novel Video Dynamics Prior (VDP) that leverages the internal statistics of a query video sequence to perform video-specific enhancement tasks. By utilizing a video's internal statistics, our approach eliminates the need for curated training data. Thus, removing the constraints with regard to encountering test time variations. Additionally, we introduce a spatial pyramid loss that enhances robustness in the temporal domain, enabling our method to handle noisy input frames encountered at test time.

VDP utilizes two fundamental properties of a video sequence: (1) Spatio-temporal patch (ST patch) recurrence and (2) Spatio-temporal consistency. ST patch recurrence is a fundamental finding [34] that reveals significant recurrence of small ST patches (e.g., 5x5x3 or 7x7x3) throughout a natural video sequence. To leverage this strong prior, we employ an LSTM-Convolution-based architecture, which is detailed in Sec. 3. Moreover, to leverage the ST patch recurrence property across scales in a video, we introduced a novel spatial pyramid loss. In Sec. 3.5, we demonstrate that our novel spatial pyramid loss helps the VDP model to be more robust to unstructured noise in both spatial and temporal domains.

Further, our approach considers the next frame prediction task to initialize our recurrent architecture. The task aims to auto-regressively predict the next frame, taking the current frame and previous frames information as context. We leveraged architecture from such a task as it promotes the spatio-temporal consistency in the processed video frames and models the complex temporal dynamics required for various downstream tasks. In our approach, we first formulate a next-frame prediction model with unknown parameters. Then depending on the downstream task, we maximize the observation likelihood with slight modification in the objective function.

This paper presents three key contributions to the field of video processing. Firstly, we introduce a novel inference time optimization technique that eliminates the need for training data collection to learn neural module weights for performing tasks. Secondly, we propose a novel spatial pyramid loss and demonstrate its effectiveness in providing robustness to spatial and temporal noise in videos. Lastly, we showcase the superior performance of our approach in multiple downstream video processing tasks, including video denoising, video frame interpolation, and video object removal. Our method outperforms current baselines and exhibits robustness in case noisy frames are provided as input for enhancement.

The rest of the paper is organized as follows. In Sec. 2, we briefly introduce the background work done in the area of video enhancement tasks. In Sec. 3, we discuss our methodology and key components of our architecture. Sec. 4-7, we briefly describe the video processing tasks and accompanying modifications to the objective function for dealing with different tasks at hand. Finally, we conclude our work in Sec. 9.

## 2  Related Works

Over the past decades, many research efforts have been made to increase the efficacy of video enhancements. Traditional methods like [33, 16, 51] utilized affine-estimation based motion approach to perform video enhancement tasks. However, these methods were very limited in their capability of learning a good representation of videos to generalize well on various scenes. All of this changed with the recent developments in deep learning techniques that considerably improve the representation of videos and deliver high performance gains on video enhancement tasks.

**Internal Learning.** In the recent past, a considerable number of deep internal learning approaches were put forth to perform image enhancement tasks using just a single test image [37, 46, 3, 10, 28, 36, 29, 48]. Such approaches utilize patch recurrence property in a natural image as a strong prior to deal with inverse problems such as image denoising [46, 10, 21], super-resolution [37, 46, 29], and object removal [46]. However, when such internal learning approaches are directly applied to videos for enhancement tasks, it creates flickering artifacts in the processed video [55, 19]. These artifacts arise as image-based methods fail to maintain a video's temporal consistency, a critical component of a video signal. Extending the deep internal-learning based approach to videos remains

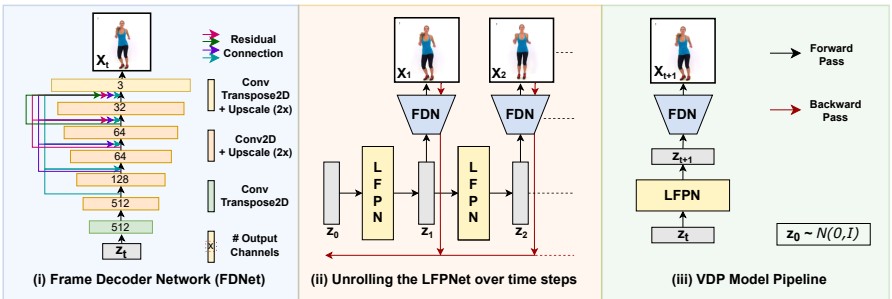

Figure 1: Model pipeline: We are given $X_1, \ldots, X_t$ frames of a person exercising (depicted in the Fig). Here, we start by sampling $z_0 \sim \mathcal{N}(0, I)$ as an initial latent representation. Our model comprises two key modules: (i) the Frame Decoder Network (FDNet) and (ii) the Latent Frame Predictor Network (LFPNet), depicted in the figure. These modules, together form the pipeline of our VDP model, as shown in (iii). We optimize the weights of these neural modules on this specific input sequence by doing backpropagation over the loss pertaining to the task at hand.

a challenging problem. There are very few works proposed in the area that deal with videos, and all of these works are tailored towards a specific video enhancement task [57, 56]. In this paper, we extend a deep internal-learning based framework to the video space that effectively explores various video enhancement tasks such as denoising, super-resolution, frame interpolation and inpainting, utilizing a single structure of our VDP model.

**Flow Guided Methods.** In the image processing tasks, most of the methods utilize the 2D CNN modules. However, if we use such models in standalone settings, we often find that they are inept at processing the additional temporal dimension present in a video sequence. In order to overcome the temporal consistency problem, some researchers have opted for an optical-flow based approach to capture the motion occurring in the videos [54, 9, 42, 26, 15, 20, 12]. Essentially, they model the flow of the video sequences by doing explicit motion estimation. Then they perform warping operations (motion compensation) to generate enhanced video frames. These optical-flow based approaches have found plenty of applications in tasks such as video denoising [54, 42], video super-resolution [54, 12, 6, 5], video object removal [11, 52], video frame interpolation [54, 2, 17, 14, 23]. However, accurate motion estimation using optical flow is inherently challenging and computationally expensive. Therefore, deep neural networks such as RAFT [44] and SPyNet [27] are employed to estimate optical flow. These networks rely on lots of training data to learn to predict the optical flow. This double reliance on data by both the video enhancement approach and optical flow prediction networks can introduce dataset biases. Further, we showcase that the flowbased models for video super-resolution and frame interpolation tasks are not robust to noise or artifacts arising from capturing the scene in low-lighting or bad lighting conditions. In our work, instead of explicitly performing motion estimation and motion compensation, we utilize LSTM-Conv decoder networks represented in Fig. 1 to capture the temporal information implicitly. Furthermore, our method is able to infer context directly from the test video sequence itself. Thus, mitigating the problems arising from dataset biases. Additionally, we empirically demonstrate that our approach is stable for tasks like super-resolution and frame interpolation, even in the presence of noisy input frames sequence.

## 3 Model Overview

Given a video sequence as context, we perform various video-specific processing tasks on this sequence of frames. In traditional settings, the training data is used to learn good representations that capture the patterns in the video sequences. However, in our case of no data, we rely on a proper choice of a deep neural network that can capture the ST patch recurrence property of a video as an informative prior for the video processing task at hand. Our model for performing all video-specific tasks is a combination of two modules. (1) The latent frame predictor network and (2) The frame decoder network as depicted in Fig. 1.(iii).

### 3.1 Latent Frame predictor Network (LFPNet)

The task of the latent frame predictor network is to maintain the spatio-temporal consistency of the video. It learns the dynamics of a video sequence in latent space, i.e., it models the temporal relations between the frames in latent space. We use LSTM cells as the building block for this module as they implicitly capture the temporal relation between the latent frames. This LFPNet module (is inspired

by [7, 39]) $g(\cdot) : \mathbb{R}^d \to \mathbb{R}^d$, has a fully-connected layer, followed by four LSTM layers with 1024 cells each, and a final output fully-connected layer. The output fully-connected layer takes the last hidden state from LSTM ($\mathbf{h}_{t+1}$) and outputs $\hat{\mathbf{z}}_{t+1}$ after $\tanh(\cdot)$ activation.

## 3.2  Frame Decoder Network (FDNet)

This network maps the frame embeddings from the latent space to the image space, i.e, $f(\cdot) : \mathbb{R}^d \to \mathbb{R}^{c \times h \times w}$. The architecture for our VDP's frame decoder network is depicted in the Fig. 1.(i). The residual skip connections in our architecture play a crucial role in the performance of our module FDNet by addressing the vanishing gradient problem and facilitating faster convergence.

## 3.3  Formulation of Video Dynamics Prior

For our proposed Video Dynamics Prior or VDP prior we only rely on the two modules namely LFPNet and FDNet. We start by sampling an initial latent noise vector $z_0 \sim \mathcal{N}(0, I)$ where $z_0 \in \mathbb{R}^D$. Here, $D$ is the dimension of latent space. For our experiments, we keep the dimension $D$ of latent space as 1024. Now, as illustrated in Fig. 1, we pass this $z_0$ embedding through the 'LFPNet' and obtain the next frame embedding $z_1$. We then pass $z_1$ through the frame decoder network to obtain the next image frame $\hat{X}_1$. Mathematically, we can write the expression for the forward pass to obtain the future frame as $\hat{X}_{t+1} = f(g(z_t))$. Where, $f : \mathbb{R}^D \to \mathbb{R}^{c \times h \times w}$ denotes the frame decoder network (§3.2) that maps the latent space to image space i.e. $X_{t+1} = f(z_{t+1})$. Function $g(\cdot)$ denotes the latent frame predictor network. This network takes the latent embedding at the current step as the input and predicts the latent embedding at the next step as the output, i.e., $z_{t+1} = g(z_t)$. Leveraging future frame prediction task helps in initializing the LFPNet in the latent space.

## 3.4  Neural Module Weights Optimization

Since we optimize the weights of neural modules (LFPNet and FDNet) using a corrupted sequence, reconstruction loss alone is not enough to find a visually plausible enhancement of the input video. Hence, we introduce a few other regularization losses.

**Reconstruction Loss:** We use a combination of L1 and perceptual loss to formulate the reconstruction loss. It is given by Eqn. 1,

$$\mathcal{L}_{\text{rec}} = \|X_{t+1} - f(g(z_t))\| + \|\phi(X_{t+1}) - \phi(f(g(z_t)))\|. \tag{1}$$

Here, $\phi(\cdot)$ in the Eqn. 1 denotes the pre-trained VGG network on ImageNet [31]. $X_t$, $\hat{X}_t$ denotes the input and reconstructed video frames at timestep t, respectively.

**Spatial Pyramid Loss:** It has been well established that small ST patches of size $5 \times 5 \times 3$ recur within and across the scales for a video. Additionally, bicubic downscaling of a video results in coarsening of the edges in the downscaled video. This results in the downscaled video to have less motion speed and less motion blur. Both of these properties are extremely useful for video enhancement tasks. Hence, to utilize such lucrative properties of a video, we introduced spatial pyramid loss. For this loss, we scale down the input image to 3 levels ($\frac{h,w}{k^2}$ where k is 2,4,8) using a downsampler and calculate the reconstruction loss for each level. It is given by Eqn. 2,

$$\mathcal{L}_{\text{spl}} = \sum_{i=2,4,8} \|d_i(X_{t+1}) - d_i(f(g(z_t)))\|. \tag{2}$$

Here, $X_{t+1}$ denotes the input video frames at timestep $t + 1$. $d_i(\cdot)$ in the Eqn. 2 denotes the downsampler network and $i$ denotes scale of downsampler. For example, $d_2(\cdot) : \mathbb{R}^{c \times h \times w} \to \mathbb{R}^{c \times h/2 \times w/2}$.

**Variation Loss:** This is a variant of Total variational loss introduced in the paper [30]. Variation loss smoothens the frames while preserving the edge details in the frame. It is given by Eqn. 3,

$$\mathcal{L}_{\text{var}}(\hat{X}_t) = \sum_{i=1}^{H-1} \sum_{j=1}^{W-1} \|\hat{x}_{ij}^t - \hat{x}_{(i-1)j}^t\| + \|\hat{x}_{ij}^t - \hat{x}_{i(j-1)}^t\|. \tag{3}$$

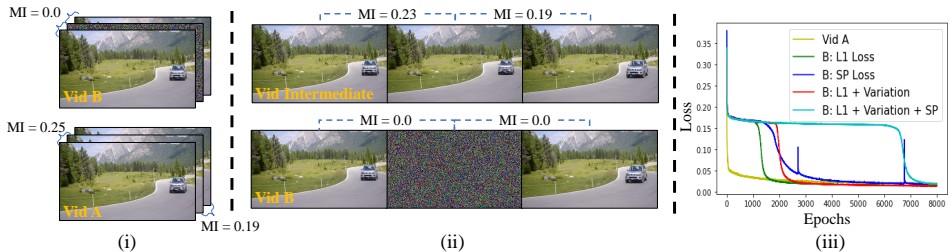

Figure 2: (i) Video B is generated by replacing the second frame of video A. (ii) Top row depicts the video obtained during the early round of optimization of the objective given by Eq. 4 for video B. Further, the average Mutual information [41] of video(intermediate) is much closer to video A. (iii) The loss plot depicts the convergence of our VDP model for in five different settings.

Here, $\hat{x}_{ij}^t$ denotes the pixel of the processed video frames ($\hat{X}_t = f(g(z_{t-1}))$) at timestep $t$ and location $\hat{X}_t[i,j]$.

**Final Loss:** We combine all the losses together into a final loss term as follows,

$$\mathcal{L}_{\text{final}} = \sum_{t=0}^{T-1} \lambda_{\text{rec}}\mathcal{L}_{\text{rec}} + \lambda_{\text{spl}}\mathcal{L}_{\text{spl}} + \lambda_{\text{var}}\mathcal{L}_{\text{var}}. \tag{4}$$

## 3.5 Robustness of Video Dynamics Prior

In this section, we explore the internal learning properties of our model. For thorough exploration, we perform various experiments to explore the spatio-temporal patch recurrence [34] in natural videos using our model. In the past, authors of the papers [10, 46] have explored the utilization of the patch recurrence property in images by the deep learning models to perform various enhancement tasks. Following these prior works, we extend such understanding to the non-trivial spatio-temporal patch (or ST-patch) recurrence phenomenon exhibited by our model.

We design a simple experiment where we have a natural video sequence A denoted as $V_a = \{X_1, X_2, X_3\}$ and depicted in Fig. 2.(i). We create another sequence $V_b$ by replacing the frame $X_2$ with a random noisy frame $X_{\text{nf}}$ resulting in $V_b = \{X_1, X_{\text{nf}}, X_3\}$. As shown in Fig. 2. (ii), the noisy frame is fairly independent of frames $X_1$ or $X_3$ hence, we get mutual information between these frames close to zero. This replacement of frame $X_2$ with $X_{nf}$ results in higher entropy [45] of the video sequence $V_b$ (larger diversity in temporal dimension $\rightarrow$ lower ST patch recurrence $\rightarrow$ lower self-similarity $\rightarrow$ higher entropy of sequence) in comparison to original sequence $V_a$(lower diversity in temporal dimension $\rightarrow$ higher ST patch recurrence $\rightarrow$ higher self-similarity $\rightarrow$ lower entropy of sequence). Hence when the VDP model is optimized over both the sequences $V_a$ and $V_b$, it requires a longer duration for loss convergence as depicted by Fig. 2.(iii). During the optimization over the video sequence $V_b$, our model obtains the sequence resembling $V_a$ (given by $V_{\text{intermediate}}$ in Fig. 2.(ii)) in early rounds of optimization, near the epochs where we observe the first dip in the convergence curve and before ending up overfitting sequence $V_b$.

We conducted such experiments on our model under five different settings. 1) Using only L1 reconstruction loss (given by Eqn. 1) on video $V_a$. 2) Using only L1 reconstruction loss on video $V_b$. 3) Using only L1 reconstruction loss and spatial pyramid loss (given by Eqn. 2) on video $V_b$. 4) Using only L1 reconstruction loss and variation loss (given by Eqn. 3) on video $V_b$. 5) Using L1 reconstruction loss, spatial pyramid loss, and variation loss on video $V_b$.

The energy plot in Fig. 2(iii) represents the trend in MSE loss for the aforementioned five settings. It can be observed that in the absence of any noise in the video signal (yellow), the model fits the input quickly. The quick convergence can be attributed to higher ST patch recurrence in video sequence A. The VDP model optimized over the video sequence $V_b$ using only the L1 reconstruction loss (green) provides some impedance to the temporal noise but quickly overfits the noise after a couple of iterations. Further, the VDP model optimized under setting 3 (red) or 4 (blue) provides higher impedance to noise than setting 2 but still converges after a few hundred epochs. However, when the VDP model is optimized using setting 5 (cyan), it provides considerably higher impedance to the noise in the temporal domain.

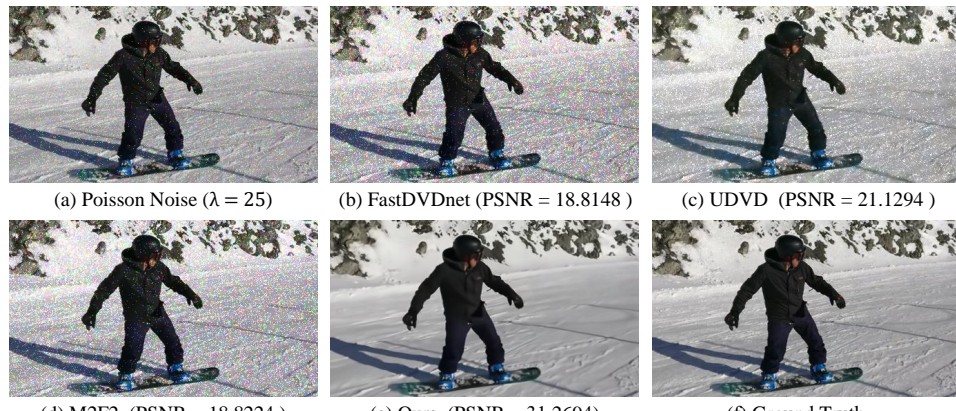

| (a) Poisson Noise (λ = 25) | (b) FastDVDnet (PSNR = 18.8148 ) | (c) UDVD  (PSNR = 21.1294 ) |
| (d) M2F2  (PSNR = 18.8224 ) | (e) Ours  (PSNR = 31.2694) | (f) Ground Truth |

Figure 3: **Video Denoising**: Frame from a video in the DAVIS dataset denoised using different approaches. (a) Frame corrupted with additive poisson noise of intensity 25 (relative to intensity range [0-255]). Figure (b) depicts the denoised frames generated by FastDVDnet [43] method. Figure (c) depicts the denoised frame generated by UDVD [35] method. Figure (d) depicts the denoised frame generated by M2F2 [8] method. Figure (e) depicts the denoised frames generated by our method. Figure (f) shows the clean ground truth frames.

Table 1: **Quantitative video denoising results:** To evaluate our method, we calculate the Avg PSNR score between the generated noise-free video and ground truth video. The $\sigma$ denotes the standard deviation for the added Gaussian noise to the frames. While the $\lambda$ denotes the intensity of additive Poisson noise. For all baselines and our method, the score is calculated on the DAVIS dataset [25]. Also, blue (**bold**) denotes the best score. Please note we use pretrained models for baselines provided by the authors. The baselines marked with ∗ denote a data-driven baseline.

| Metric | Additive Noise Type | Noise Level | M2F2 [8]∗ | FastDVDnet [43]∗ | UDVD [35]∗ | Ours |
|---|---|---|---|---|---|---|
| PSNR | Gaussian | $(\sigma = 20)$ | 34.89 | **35.77** | 35.12 | 35.74 |
| PSNR | Gaussian | $(\sigma = 30)$ | 33.21 | 34.04 | 33.92 | **34.07** |
| PSNR | Poisson | $(\lambda = 25)$ | 18.19 | 19.02 | 22.08 | **31.96** |
| PSNR | Poisson | $(\lambda = 30)$ | 17.12 | 18.03 | 20.98 | **30.07** |

We observe a similar phenomenon when optimizing our model with a larger void filled with noisy frames. Additionally, when we optimize our VDP model over a noisy video (here, every frame in the video sequence consists of Gaussian noise), we do not see convergence at all. We attribute this behavior of our VDP model to promoting self-similarity of the st-patches over the output video sequence. Moreover, incorporating the spatial pyramid loss introduces frame matching at multiple levels of coarse scale. This encourages a more structured signal in the output and provides a higher impedance to the unstructured noise. We performed the same experiment over 250 different video sequences (randomly sampled from different datasets like Vimeo-90K, DAVIS, and UCF101) and came up with similar findings as given by Fig. 2.(iii).

## 4   Video Denoising

In this section, we present our VDP model utilized for the video denoising task. A noisy video can be defined as the combination of a clean video signal and additive noise. As shown in the previous section, our VDP model exhibits higher impedance to the noise signal and lower impedance to the clean video signal. Exploiting this property, we propose a denoiser for video sequences. Unlike traditional approaches, our denoiser does not require data collection, as we demonstrate that the final loss function (Eqn. 4) alone is sufficient for effective video denoising.

We optimize our model parameters on the noisy test-time video sequence using the loss function given by Eqn. 4. We then use these optimized LFPNet & FDNet modules for generating high-quality clean video frames. The new enhanced video frames for each timestamp can be obtained by $\hat{X}_{t+1} = f(g(z_t))$.

The presence of noise in a video can be attributed to various factors, such as non-ideal video capture settings or lossy video streaming over the internet. While curating training datasets for baselines, it

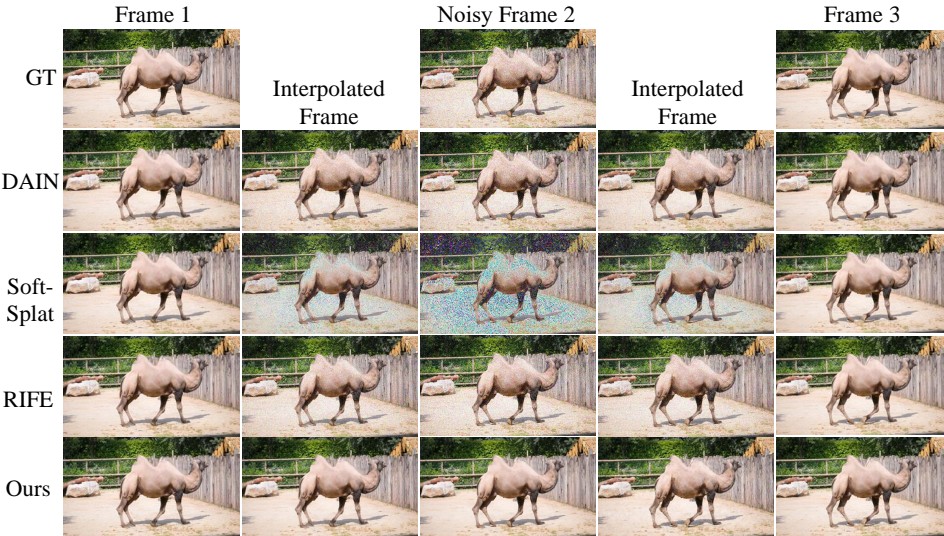

Figure 4: **Video Frame Interpolation**: In order to compare our method to existing baseline methods, we conducted visual comparisons using three frames for each method. We introduced Gaussian noise ($\sigma = 15$) in the second frame and tasked all methods with interpolating frames between them. It can be observed that our method not only denoises the second noisy input frame but also generates denoised intermediate frames. This is in contrast to other baseline methods that interpolate noise along with the clean signal.

Table 2: **Quantitative video frame interpolation results:** For the evaluation of our method we calculate the Avg PSNR and SSIM score between the generated interpolated video frames and ground truth video. For all baselines and our method, the score is calculated on the UCF101 dataset [40]. The color blue (**bold**) denotes the best performance. Please note we use pretrained models for baselines provided by the authors. The baselines marked with ∗ denote a data-driven baseline.

| Metrics | # Noisy Frames | Noise level | ToFlow [54]* | DAIN [2]* | SoftSplat [23]* | RIFE [14]* | Ours |
|---------|----------------|-------------|--------------|-----------|-----------------|------------|------|
| PSNR | None | None | 34.68 | 35.00 | 35.39 | **35.41** | **35.41** |
| SSIM | None | None | 0.9677 | 0.968 | **0.970** | **0.970** | **0.970** |
| PSNR | 2nd Frame | $\sigma = 15$ | 20.30 | 20.87 | 18.25 | 20.22 | **34.92** |
| SSIM | 2nd Frame | $\sigma = 15$ | 0.514 | 0.521 | 0.4012 | 0.512 | **0.918** |

is common to assume that the additive noise follows a Gaussian distribution. However, at test time, when this additive noise is drawn from a Poisson distribution, the performance of baseline models deteriorates significantly. In contrast, our VDP model, optimized on the test sequence by minimizing the loss given in Eqn. 4, achieves highly effective denoising results. Fig. 3 provides a qualitative comparison between our method and the baselines, clearly demonstrating the superior performance of our approach.

To further evaluate the effectiveness of our method, we conduct experiments on the DAVIS dataset using different noise settings. We compare our model against state-of-the-art baselines and present the quantitative evaluation in Table. 1. The results highlight the superiority of our method in video denoising tasks, showcasing its robustness in handling diverse noise distributions. We encourage readers to refer to our supplementary material for additional video results.

## 5   Video Frame Interpolation

Frame interpolation is a challenging task that involves generating intermediate frames between the given frames in a video sequence. The primary objective is to synthesize high-quality and realistic frames that accurately capture the spatio-temporal coherence of the original video. However, existing baseline methods for frame interpolation assume that the input video consists of high-quality frames without any noise. In real-world settings, video sequences often contain noise due to non-ideal capturing conditions, which poses challenges and diminishes the performance of these baseline methods. Such limitations are addressed by our VDP model.

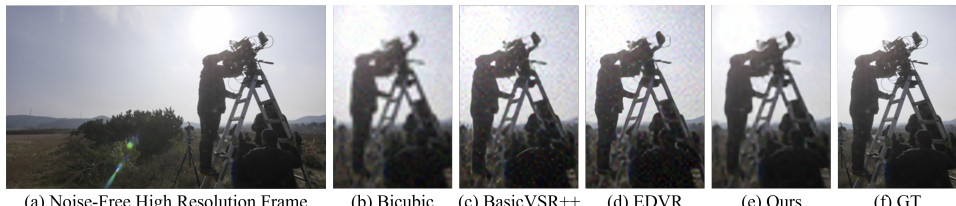

| (a) Noise-Free High Resolution Frame | (b) Bicubic | (c) BasicVSR++ | (d) EDVR | (e) Ours | (f) GT |

Figure 5: **Video Super Resolution**: Comparison of results of the 'Cameraman' sequence from VIMEO-90K-T dataset. Gaussian noise of std $\sigma = 5$ is added to the lower-resolution input frames. (a) High-resolution noise-free ground truth frame. (b) Patch from higher resolution frame generated by BiCubic method. (c) Patch from higher resolution generated by BasicVSR++ [5]. (d) Patch from higher resolution frame generated by EDVR [49] method. (e) Patch from the higher resolution by Ours VDP method. (f) Patch from higher resolution ground truth frame. Note the noise suppression and details of the camera in the patch generated by our method compared to baselines.

Table 3: **Quantitative video super-resolution results:** For the evaluation of our method, we calculate the Avg PSNR and SSIM scores between the generated super-resolution video and ground truth video for all baselines along with our method. The score is calculated on the standard VIMEO-90K-T dataset. Best results in the table are shown in blue (**bold**). Please note we use pretrained models for baselines provided by the authors. The baselines marked with $*$ denote a data-driven baseline.

| Metrics | Noisy Frames | Noisy Intensity | BiCubic | ToFlow [54]$^*$ | EDVR [49]$^*$ | BasicVSR++ [5]$^*$ | Ours |
|---|---|---|---|---|---|---|---|
| PSNR | ✗ | - | 31.12 | 33.20 | 35.79 | **35.95** | 35.70 |
| SSIM | ✗ | - | 0.870 | 0.920 | 0.937 | **0.940** | 0.936 |
| PSNR | ✓ | $\sigma = 5$ | 29.90 | 30.20 | 32.02 | 31.58 | **33.87** |
| SSIM | ✓ | $\sigma = 5$ | 0.796 | 0.712 | 0.758 | 0.720 | **0.878** |

For this task, we utilize the latent embeddings to perform the frame interpolation. First, we optimize our VDP model parameters using the loss function given by Eqn. 4. Now, to generate intermediate frames, we linearly interpolate the latent embeddings of the video using the equation.

$$X_{\text{intermediate}} = f(\alpha z_{t+1} + (1 - \alpha)z_t) \quad \forall \alpha \in (0, 1). \tag{5}$$

Here, $X_{\text{intermediate}}$ is the interpolated frame between the frames $X_t$ and $X_{t+1}$. For example, to perform a 4x frame interpolation, we select $\alpha = [0.25, 0.5, 0.75]$ in the Eqn. 5. To evaluate the performance of our method, we conduct experiments on the UCF101 (triplet) dataset [2, 40] under two conditions: 1) ideal triplet input sequence, 2) triplet input sequence containing one noisy frame. We make a qualitative comparison between our method and the baselines depicted in Fig. 4. It can be observed from Fig. 4 that our method is more robust to noisy input and outputs the crisp intermediate frames without noise, unlike baseline methods DAIN [2], SoftSplat [23] and RIFE [14].

Table. 6 represents the quantitative evaluation of our approach on the UCF-101 dataset. It can be observed from the table that our method outperforms the existing baselines on PSNR and SSIM metrics in the video frame interpolation task, further validating the effectiveness and suitability of our approach for this task.

## 6 Video Super Resolution

Given our ever-increasing screen resolutions, there is a great demand for higher-resolution video content. In this task, we are given a lower-resolution video sequence as input. We are tasked to change this input to a higher-resolution video sequence.

For this task, we modify the loss objective given by Eqn. 4 as follows,

$$\arg\min_{f,g} \sum_{t=0}^{T-1} \lambda_{\text{rec}}\mathcal{L}_{\text{rec}}(X_{t+1}, d_{\text{sc}}(f(g(z_t)))) + \lambda_{\text{spl}}\mathcal{L}_{\text{spl}}(X_{t+1}, d_{\text{sc}}(f(g(z_t)))) + \lambda_{\text{var}}\mathcal{L}_{\text{var}}(f(g(z_t))). \tag{6}$$

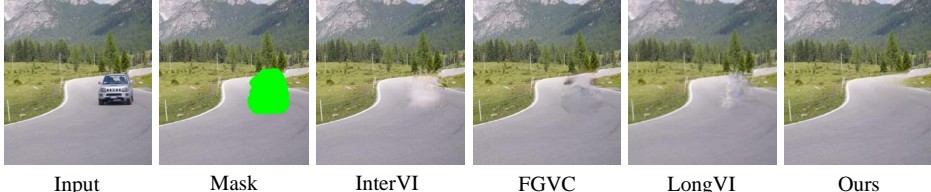

| Input | Mask | InterVI | FGVC | LongVI | Ours |

Figure 6: **Video Object Removal**. In the above figure, we describe a video object removal task. The input in the figure represents the original video sequence frame without any alterations. The InterVI [56], FGVC [11], and LongVI [24] in the figure represent the frames generated by removing the object "car" from the video sequence. The rightmost frame depicts the removal of the object from the video frame by our method. It can be observed that our method do not create hallucinations.

Table 4: **Quantitative Object Removal results:** We compare the average PSNR and SSIM scores of our method with the existing baselines on the DAVIS dataset. The best performance is denoted by the score in blue (**bold**). Please note we use pretrained models for baselines provided by the authors. The baselines marked with $*$ denote a data-driven baseline.

| Metrics | DIP [46] | DFGVI [53]$^*$ | FGVC [11]$^*$ | InterVI [56] | LongVI [24] | Ours |
| --- | --- | --- | --- | --- | --- | --- |
| PSNR | 26.03 | 27.57 | 31.92 | 30.86 | 31.80 | **32.06** |
| SSIM | 0.921 | 0.9289 | 0.9499 | 0.9350 | 0.9457 | **0.9512** |

The modified objective for the video super-resolution task is given by Eqn. 6. Here, function $d_{sc}(\cdot)$ denotes the downsampling function. For example, to perform 4x super-resolution, we would define downsampling function as $d_{sc} : \mathbb{R}^{c \times h \times w} \rightarrow \mathbb{R}^{c \times (h/4) \times (w/4)}$.

We evaluate our method for the video super-resolution task on the VIMEO90-K-T dataset and contrast our model against the state-of-the-art baseline methods. We evaluate our method on two different settings: 1) When sharp lower-resolution frames are provided, 2) When noisy lower-resolution frames are provided. We perform a $4\times$ super resolution of frames in the video sequences. We make a qualitative comparison between our method and the baselines for the setting when noisy frames are given as input frames, depicted in Fig. 7. It can be observed from Fig. 7 that our method outputs better and noise-free higher resolution frames with more details in comparison to the baseline methods. Table. 5 represents the quantitative evaluation of our approach. It can be observed from the table that there is a significant drop in the performance of baselines to our method when a slight amount of Gaussian noise is added to the input frames. We demonstrate empirically that our method is much more robust to noise as compared to other baseline methods.

## 7 Video Object Removal

Unwanted items in a video can grab the viewers' attention from the main content. In this task, we remove such objects from the original video stream. For this task, we assume a mask $M = \{m_t\}_1^T \forall m_t \in \{0,1\}^{c \times w \times h}$ is provided for each frame to remove an object from the video sequence. To perform the removal of an object, we modify the objective function given by Eqn. 4 as follows,

$$\arg\min_{f,g} \sum_{t=1}^{T} \lambda_{rec}\mathcal{L}_{rec}(m_t X_t, m_t f(g(z_{t-1}))) + \lambda_{spl}\mathcal{L}_{spl}(m_t X_t, m_t f(g(z_{t-1}))) + \lambda_{var}\mathcal{L}_{var}(f(g(z_{t-1}))).$$

(7)

We evaluate our method for the video object removal task on the DAVIS dataset. A qualitative comparison between our method and the baselines can be seen in Fig. 6. It can be observed from Fig. 6 that our method performs better than the existing methods in removing the object from the video without creating any artifacts in the processed videos.

Table. 4 represents the quantitative evaluation of our approach. We perform the evaluation based on the stationary mask setting. In this setting, a stationary mask is applied to all frames of the video sequence. This setting helps us evaluate the model's ability to generate temporally coherent frames even if some information in the frames is consistently missing.

## 8   Limitations

One severe limitation of our approach is that it is an offline approach, i.e., it can not be used in real-time to perform video enhancements. One way to rectify this limitation is to work on reducing the parameters for the frame decoder network. One suggestion is modifying the residual skip connection from concatenation-based connections to additive connections. This would drastically reduce the number of parameters to be trained with minimal effects on the efficacy of the model.

The second limitation of our approach is that it is a recurrent approach and can not be scaled using multiple GPUs. This issue can be alleviated using a temporal attention-based approach that can have long-term memory and is scalable across multiple GPUs.

Our approach, in its current stage, requires no data and is only able to perform low-level vision tasks. However, to perform high-level vision tasks such as performing video segmentation or video manipulation, we conjecture that external knowledge would be required, like a collection of external datasets or a pretrained model on a such external dataset to aid our VDP method.

## 9   Conclusion

We conclude in this work that a video-specific approach is a way to go for a considerable number of video processing tasks. Our VDP method not only performed better than the task-specific approaches but also it is not hamstrung by the limited availability of either the computing power or the training data. We demonstrate that the introduction of a spatial pyramid loss enhances the robustness of our proposed VDP method. Leveraging this loss, we were able to achieve state-of-the-art results in all enhancement tasks, even in the presence of noise in the input frames. One limitation of our method is the long processing time compared to other methods at the test time inference. Nevertheless, it is important to note that we do not require training as the other baseline methods (requires multiple GPU hrs if not days). In the future, we will focus on improving efficiency to shorten the processing time for increased practical utility. One great advantage of our approach is removing the data collection requirement for such processing tasks. We hope that this step towards a data-free approach will reduce the massive data mining across the internet and provide better data privacy measures.

## 10   Broader Impact

Our approach offers a significant advantage by eliminating the need for data collection in processing tasks. This shift towards a data-free approach has the potential to mitigate the extensive data mining activities conducted across the internet, thereby addressing concerns related to data privacy. By reducing the reliance on data collection, our approach promotes more ethical and responsible use of technology.

However, it is crucial to acknowledge that our approach also has some limitations. We believe that addressing these challenges through ongoing research and development will contribute to the continuous improvement and refinement of data-free approaches, thereby enhancing their broader impact.

## 11   Acknowledgement

This project was partially funded by the DARPA SAIL-ON (W911NF2020009) program and an independent grant from Meta AI.

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

# Contents

## A Comparison with cascaded models

Existing video enhancement approaches rely on clean and sharp input frames to perform the task at hand. However, this is not always the case depending on the lighting condition of scene capture or compressing a video stream to send it over the internet. All of these introduce noise in the clean video signal. Applying video enhancement approaches in such conditions results in sub-optimal enhancements, as demonstrated in the main paper. One other way of dealing with such conditions is the use of a two-stage cascaded approach. For example, we have to provide super-resolute frames for a noisy video sequence. We can get the solution by first denoising the frames and then processing these with super-resolution approaches. However, this also results in error propagation from one stage to another and results in artifacts.

To evaluate the aforementioned reasoning, we perform a two-stage cascaded approach for video super-resolution and frame interpolation tasks. In this cascaded approach, we apply the frame denoiser first and then apply the VSR or VFI model. For video denoiser we use the FastDVDnet [43]. Table. 5 and 6 describe the quantitative evaluation of baselines(cascaded approach) with our VDP model for super-resolution and frame interpolation tasks. It can be observed that there is a deterioration in results for the super-resolution task for the cascaded approach than directly applying the baseline. This is primarily due to the fact that when the denoiser is applied in the first stage, it washes away important details from the lower-resolution frames. When these frames are passed through the video super-resolution baselines, the error from the first stage is propagated further. A qualitative visualization of this is depicted in Fig. 7. However, for the frame interpolation task, the case is reversed, and cascaded baseline approaches yield better results than only baseline approaches. It can be seen from Table. 6 that the cascaded baselines perform better than only baseline approaches but still perform worse than our VDP approach.

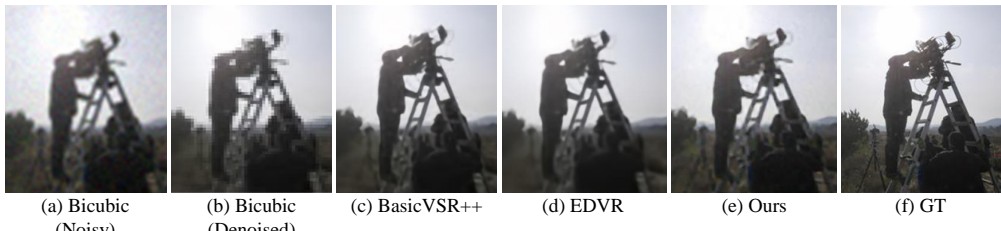

| (a) Bicubic | (b) Bicubic | (c) BasicVSR++ | (d) EDVR | (e) Ours | (f) GT |
| (Noisy) | (Denoised) | | | | |

Figure 7: **Video Super Resolution**: Comparison of results of the 'Cameraman' sequence from VIMEO-90K-T dataset. Gaussian noise of std $\sigma = 5$ is added to the lower-resolution input frames. (a) Bicubic extrapolation of noisy low-resolution frame (b) Bicubic extrapolation of denoised low-resolution frame. (c) Patch from higher resolution generated by BasicVSR++ [5] over the denoised low-resolution frame. (d) Patch from higher resolution frame generated by EDVR [49] over the denoised low-resolution frame. (e) Patch from the higher resolution by Ours VDP method. (f) Patch from higher resolution ground truth frame. We use standard video denoiser FastDVDNet [43] to denoise the input low-resolution video frames. Based on the figure, it appears that the denoiser used in the first stage of the model removes vital information from the lower-resolution frames, which leads to errors in the second stage. Our VDP model, on the other hand, preserves more details compared to the baselines.

Table 5: **Quantitative cascaded video super-resolution($4\times$) results:** For the evaluation of our method, we calculate the Avg PSNR and SSIM scores between the generated super-resolution video and ground truth video for all baselines along with our method. The score is calculated on the standard VIMEO-90K-T dataset. Best results in the table are shown in blue (**bold**). Please note we use VSR baselines by cascading it with the video denoiser model FastDVDnet [43]. The baselines marked with $*$ denote a data-driven baseline.

| Metrics | Noisy Frames | Noisy Intensity | BiCubic | EDVR [49]$^*$ | BasicVSR++ [5]$^*$ | Ours |
|---|---|---|---|---|---|---|
| PSNR | ✓ | $\sigma = 5$ | 27.78 | 31.56 | 31.03 | **33.87** |
| SSIM | ✓ | $\sigma = 5$ | 0.806 | 0.857 | 0.853 | **0.878** |

Also, we have shown in the main paper that denoisers like FastDVDnet [43] are also prone to variations in noise distribution. This would hurt the efficacy of the cascaded models if the noise is coming from a distribution other than the Gaussian distribution.

Table 6: **Quantitative cascaded video frame interpolation results:** For the evaluation of our method, we calculate the Avg PSNR and SSIM score between the generated interpolated video frames and ground truth video. For all baselines and our method, the score is calculated on the UCF101 triplet dataset. Please note we use VFI baselines by cascading it with the video denoiser model FastDVDnet [43]. The color blue (**bold**) denotes the best performance. The baselines marked with $*$ denote a data-driven baseline.

| Metrics | # Noisy Frames | Noise level | ToFlow [54]$^*$ | DAIN [2]$^*$ | SoftSplat [23]$^*$ | RIFE [14]$^*$ | Ours |
|---|---|---|---|---|---|---|---|
| PSNR | 2nd Frame | $\sigma = 15$ | 33.31 | 33.51 | 34.20 | 33.95 | **34.92** |
| SSIM | 2nd Frame | $\sigma = 15$ | 0.882 | 0.891 | 0.898 | 0.892 | **0.918** |

# B  Ablation study

We performed ablation experiments to study the effect of losses for our video dynamics prior. We report the quantitative results for all the enhancement tasks in Table. 7. It can be observed from the denoising section of the table that spatial pyramid loss $\mathcal{L}_{spl}$ (given by Eqn. 2) is very important to provide robustness to both the spatial and temporal domains. Additionally, it can be observed that variation loss $\mathcal{L}_{var}$ is very important for tasks like super-resolution, frame interpolation, and object removal. In Fig. 8-11, we show a qualitative comparison between VDP model optimized with different subsets of losses.

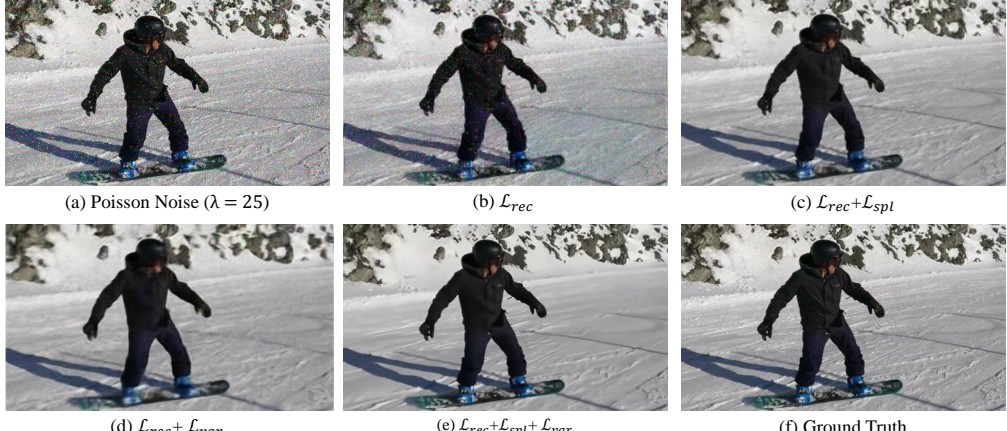

| (a) Poisson Noise ($\lambda = 25$) | (b) $\mathcal{L}_{rec}$ | (c) $\mathcal{L}_{rec}+\mathcal{L}_{spl}$ |
| (d) $\mathcal{L}_{rec}+\mathcal{L}_{var}$ | (e) $\mathcal{L}_{rec}+\mathcal{L}_{spl}+\mathcal{L}_{var}$ | (f) Ground Truth |

Figure 8: **Video Denoising (Ablation)**: Frame from a video in the DAVIS dataset denoised using loss functions. (a) Frame corrupted with additive poisson noise of intensity 25 (relative to intensity range [0-255]). Figure (b) depicts the denoised frames generated by only $\mathcal{L}_{rec}$. Figure (c) depicts the denoised frame generated by $\mathcal{L}_{rec} + \mathcal{L}_{spl}$. Figure (d) depicts the denoised frame generated by $\mathcal{L}_{rec} + \mathcal{L}_{var}$. Figure (e) depicts the denoised frames generated by $\mathcal{L}_{rec} + \mathcal{L}_{spl} + \mathcal{L}_{var}$. Figure (f) shows the clean ground truth frames.

**Video denoising:** It can be observed from Fig. 8 that only using the reconstruction loss $\mathcal{L}_{rec}$ results in very suboptimal denoising. Optimizing for a high number of epochs using only $\mathcal{L}_{rec}$ results in the total reconstruction of the noisy signal. Adding the variation loss $\mathcal{L}_{var}$ to $\mathcal{L}_{rec}$ results in smoothening of the video frame as can be observed from the Fig. 8. However, adding the spatial pyramid loss $\mathcal{L}_{spl}$ to $\mathcal{L}_{rec}$ results in a very strong denoiser. This can also be observed in the quantitative evaluation given by Table. 7.

**Video Super Resolution:** It can be observed from Fig. 9 that only using the reconstruction loss $\mathcal{L}_{rec}$ results in suboptimal quality of super resolute frames with edginess in the texture. The addition of the spatial pyramid loss $\mathcal{L}_{spl}$ to $\mathcal{L}_{rec}$ results in better texture in the high-resolution frames, yet some edgyness in the texture remains. However, the addition of the variation loss $\mathcal{L}_{var}$ to $\mathcal{L}_{spl} + \mathcal{L}_{rec}$ results in smoothening of the texture while retaining the edges. This can also be observed in the quantitative evaluation given by Table. 7.

**Video Frame Interpolation:** It can be observed from Fig. 10 that only using the reconstruction loss $\mathcal{L}_{rec}$ results in suboptimal quality of interpolated frames with rough textures and missing details. The addition of the spatial pyramid loss $\mathcal{L}_{spl}$ to $\mathcal{L}_{rec}$ results in better texture and increased details in the interpolated frames however, some edginess in the texture remains. However, the addition of the variation loss $\mathcal{L}_{var}$ to $\mathcal{L}_{spl} + \mathcal{L}_{rec}$ results in smoothening of the texture while retaining the details. This can also be observed in the quantitative evaluation given by Table. 7.

**Video Object Removal:** It can be observed from Fig. 11 that only using the reconstruction loss $\mathcal{L}_{rec}$ results in suboptimal quality of interpolated region in the frames with rough textures and color jitter artifacts. The addition of the spatial pyramid loss $\mathcal{L}_{spl}$ to $\mathcal{L}_{rec}$ results in better texture and reduced jittering artifacts in the interpolated region in the frames. However, the addition of the variation loss $\mathcal{L}_{var}$ to $\mathcal{L}_{spl} + \mathcal{L}_{rec}$ results in smoothening of the texture and merging of the inpainted region with the background. This can also be observed in the quantitative evaluation given by Table. 7.

## C  Processing time

We would like to point out here that if we take cumulative training and inference time, then our baseline would come out to be a better bargain in many scenarios, as depicted by Table. 8. To put it in perspective, a data-heavy baseline requires an ample amount of time to train, even when provided with perfect hyperparameters, making exploring different architectures quite expensive. In addition, the training dataset is limited in terms of representing real-world scenarios; this would further increase

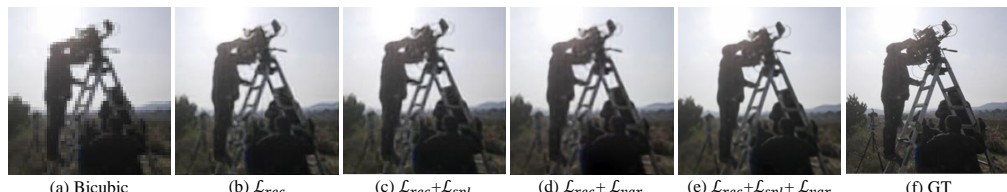

| (a) Bicubic | (b) $\mathcal{L}_{rec}$ | (c) $\mathcal{L}_{rec}+\mathcal{L}_{spl}$ | (d) $\mathcal{L}_{rec}+\mathcal{L}_{var}$ | (e) $\mathcal{L}_{rec}+\mathcal{L}_{spl}+\mathcal{L}_{var}$ | (f) GT |

Figure 9: **Video Super Resolution (Ablation)**: Comparison of results of the 'Cameraman' sequence from VIMEO-90K-T dataset. (a) Bicubic extrapolation of low-resolution frame. (b) Patch from higher resolution generated by utilizing only the reconstruction loss given by Eqn. 1. (c) Patch from higher resolution frame generated by $\mathcal{L}_{rec} + \mathcal{L}_{spl}$ (d) Patch from the higher resolution frame generated by $\mathcal{L}_{rec} + \mathcal{L}_{var}$. (e) Patch from the higher resolution frame generated by $\mathcal{L}_{rec} + \mathcal{L}_{spl} + \mathcal{L}_{var}$ (f) Patch from higher resolution ground truth frame.

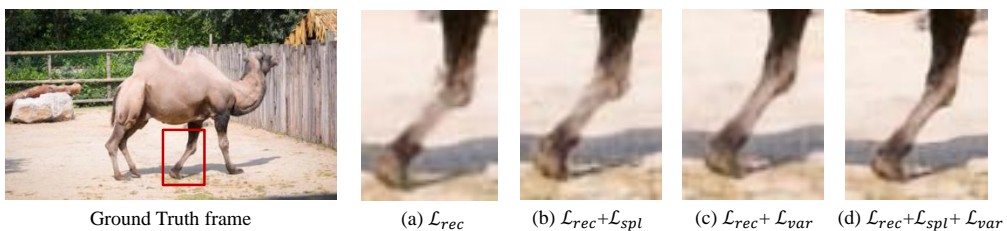

| Ground Truth frame | (a) $\mathcal{L}_{rec}$ | (b) $\mathcal{L}_{rec}+\mathcal{L}_{spl}$ | (c) $\mathcal{L}_{rec}+\mathcal{L}_{var}$ | (d) $\mathcal{L}_{rec}+\mathcal{L}_{spl}+\mathcal{L}_{var}$ |

Figure 10: **Video Frame Interpolation (Ablation)**: In order to compare our method with different loss functions, we conducted a visual comparison. (a) Patch generated by utilizing only the reconstruction loss given by Eqn. 1. (b) Patch generated by $\mathcal{L}_{rec} + \mathcal{L}_{spl}$ (c) Patch generated by $\mathcal{L}_{rec} + \mathcal{L}_{var}$. (d) Patch generated by $\mathcal{L}_{rec} + \mathcal{L}_{spl} + \mathcal{L}_{var}$.

the spending time and effort in fine-tuning the network every time for a new setting. Please note that the inference time for our method has been calibrated on one Nvidia A6000 GPU.

## D  Optimization Details

To optimize our model, we use a single Nvidia A6000 GPU with 48G memory to process a single video at a time with 15-frame sequences of size 448x256. We optimize the modules LFPNet and FDNet weights using the entire test sequence with the *Adam optimizer* at a learning rate in the range of [0.0002, 0.002].

**Video Denoising:** For getting a good performance on the denoising task, we utilize the following weights for the different losses; $\lambda_{rec} = 1$, $\lambda_{spl} = 0.0001$ and $\lambda_{var} = 0.0001$. We observed that the performance of our VDP denoiser plateaued after 3600 epochs.

**Video Frame Interpolation:** For getting a good performance on the frame interpolation task, we utilize the following weights for the different losses; $\lambda_{rec} = 1$, $\lambda_{spl} = 0.0001$ and $\lambda_{var} = 0.0001$. We observed that the performance of our VDP frame interpolation model plateaued after 1800 epochs.

Table 7: **Quantitative ablations results:** To evaluate our method, we calculate the Avg PSNR score between the generated noise-free video and ground truth video. blue (**bold**) denotes the best score.

| Task | Dataset | Metric | Additive Noise Type | Noise Level | $\mathcal{L}_{rec}$ | $\mathcal{L}_{rec} + \mathcal{L}_{var}$ | $\mathcal{L}_{rec} + \mathcal{L}_{spl}$ | $\mathcal{L}_{rec} + \mathcal{L}_{var} + \mathcal{L}_{spl}$ |
|---|---|---|---|---|---|---|---|---|
| Denoising | DAVIS | PSNR | Poisson | $(\lambda = 25)$ | 20.58 | 24.63 | 30.14 | **31.96** |
| | | PSNR | Poisson | $(\lambda = 30)$ | 20.11 | 23.82 | 29.81 | **30.07** |
| Super-Resolution | Vimeo-90KT | PSNR | ✗ | ✗ | 31.23 | 33.86 | 32.18 | **35.70** |
| | | SSIM | ✗ | ✗ | 0.890 | 0.924 | 0.915 | **0.936** |
| Frame-interpolation | UCF-101 | PSNR | ✗ | ✗ | 33.29 | 34.13 | 33.88 | **35.41** |
| | | SSIM | ✗ | ✗ | 0.943 | 0.945 | 0.963 | **0.970** |
| Object Removal | DAVIS | PSNR | ✗ | ✗ | 29.31 | 31.78 | 30.12 | **32.06** |
| | | SSIM | ✗ | ✗ | 0.9125 | 0.9451 | 0.9352 | **0.9512** |

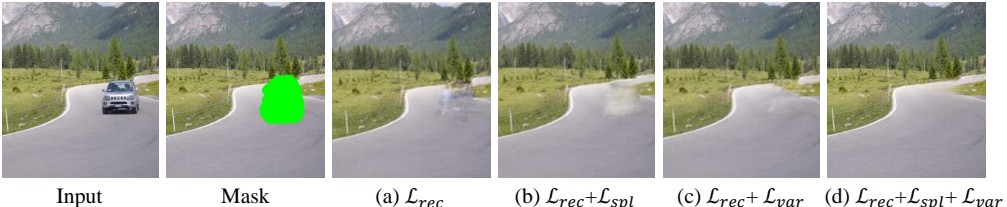

| Input | Mask | (a) $\mathcal{L}_{rec}$ | (b) $\mathcal{L}_{rec}+\mathcal{L}_{spl}$ | (c) $\mathcal{L}_{rec}+\mathcal{L}_{var}$ | (d) $\mathcal{L}_{rec}+\mathcal{L}_{spl}+\mathcal{L}_{var}$ |

Figure 11: **Video Object Removal (Ablation)**. In the above figure, we describe a video object removal task. The GT in the figure represents the original video sequence frame without any alterations. (a) Patch generated by utilizing only the reconstruction loss given by Eqn. 1. (b) Patch generated utilizing the loss given by $\mathcal{L}_{rec} + \mathcal{L}_{spl}$ (c) Patch generated by utilizing the loss given by $\mathcal{L}_{rec} + \mathcal{L}_{var}$. (d) Patch generated by utilizing the loss given by $\mathcal{L}_{rec} + \mathcal{L}_{spl} + \mathcal{L}_{var}$.

Table 8: Training and inference time comparison with the best baselines. Note that the inference time in the table is given as the time required to process per frame of a video.

| Task | Dataset size | Resolution | Ours (Train + Infer) | Best Baseline (Train + Infer) |
|------|--------------|------------|----------------------|-------------------------------|
| Super-Resolution | 38,990 (7 fr/v) | $112 \times 64 \rightarrow 448 \times 256$ | 0 + 15s | 300 hrs + 1s |
| Denoise | 50 seq (3450 fr) | $448 \times 256$ | 0 + 12s | 200 hrs + 0.1s |
| Object Removal | 50 seq (3450 fr) | $448 \times 256$ | 0 + 8.5s | 160 hrs + 8.3s |
| Frame Interpolation | 3,782 triplets | $448 \times 256$ | 0 + 10s | 120hrs + 0.08s |

**Video Super-Resolution :** For getting a good performance on the VSR task, we utilize the following weights for the different losses; $\lambda_{rec} = 1$, $\lambda_{spl} = 0.01$ and $\lambda_{var} = 0.0001$. We observed that the performance of our VDP SR model plateaued after 4200 epochs.

**Video Object removal:** For getting a good performance on the object removal task, we utilize the following weights for the different losses; $\lambda_{rec} = 1$, $\lambda_{spl} = 0.01$ and $\lambda_{var} = 0.0001$. We observed that the performance of our VDP object removal model plateaued after 1800 epochs.

Additionally, we conducted our experiments using multiple random initializations of FDNet and LFPNet modules and observed consistent quantitative results across all runs.

# E  Additional Details - FDNet

Each convolution layer is followed by batch normalization and a leaky rectified linear unit (LeakyReLU; negative slope = 0.2), except for the last layer of the decoder. In the last layer of the decoder, we use the sigmoid activation function. We utilize Pytorch for our implementation.

# F  Dataset Descriptions

**Vimeo-90K-T [54]** This dataset contains 7824 short clips downloaded from 'vimeo.com'. Each clip only contains 7 frames per clip. We used this dataset for a video super-resolution task. The lower resolution input frames have a size of $3 \times 112 \times 64$ while the higher resolution frames have a size of $3 \times 448 \times 256$.

**DAVIS [25]** This dataset contains 50 curated video clips, and each clip contains a unique object in the clip. The dataset has masking annotation present for this object for each clip. Each frame in each clip contains 1 mask annotation. There is 3450 total number of frames in this dataset. We used this dataset for video denoising and object removal task. We resized the resolution of each clip for the aforementioned tasks. The resolution of input frames for the task was of the size $3 \times 448 \times 256$.

**UCF [40]** The UCF101 dataset contains videos with a large variety of human actions. We utilized this dataset for the video frame interpolation task. We used this dataset under standard settings put forth by paper [2]. There are 379 triplets with a resolution of $256 \times 256$ pixels.

Table 9: We include LPIPS and FVD [47] metrics for completeness of quantitative evaluation of enhancement tasks (video denoising, object removal, and video interpolation). FVD is a deep neural network based metric that evaluates both the spatial and temporal quality of the processed video with the ground truth video. FVD requires minimum of 16 frames in a sequence for evaluation. Hence, we were unable to evaluate FVD for task like VFI and VSR on UCF101 triplet and Vimeo90KT datasets.

| Task | Method | Dataset | Noisy Frames | Noise Intensity | FVD↓ | LPIPS↓ | PSNR↑ | SSIM↑ |
|------|--------|---------|--------------|-----------------|------|--------|-------|-------|
| Denoise | FastDVDnet | DAVIS | ✓ | $\lambda = 25$ | 1743 | 0.65 | 19.02 | - |
| (Poisson) | **Ours** | | | | **54** | **0.0056** | **31.96** | - |
| Object | FGVC | DAVIS | ✗ | - | 580 | 0.0556 | 31.92 | 0.9499 |
| Removal | **Ours** | | | | **445** | **0.0408** | **32.06** | **0.9512** |
| Inter- | RIFE | UCF101 | $2^{nd}$ Frame | $\sigma = 15$ | - | 0.412 | 20.22 | 0.512 |
| -polation | SoftSplat-$\mathcal{L}_F$ | | | | - | 0.624 | 18.25 | 0.401 |
| | **Ours** | | | | - | **0.102** | **34.92** | **0.918** |
| VSR | EDVR | Vimeo90KT | ✓ | $\sigma = 5$ | - | 0.5051 | 32.02 | 0.758 |
| | BasicVSR++ | | | | - | 0.5263 | 31.58 | 0.720 |
| | **Ours** | | | | | **0.181** | **33.87** | **0.878** |

