# OpenReview forum: "Video Dynamics Prior: An Internal Learning Approach for Robust Video Enhancements"
_NeurIPS.cc/2023/Conference — NeurIPS 2023 poster_

### Official Review · Reviewer_fnUo · 2023-06-27

**Soundness:** 3 good
**Presentation:** 3 good
**Contribution:** 2 fair
**Rating:** 3
**Confidence:** 5

**Summary:**

The paper presents a framework for various low-level vision tasks, such as denoising, object removal, frame interpolation, and super-resolution. What sets this framework apart is that it does not require any external training data. Instead, it directly leverages the internal statistics of videos at test-time to optimize neural network modules to perform the target task. Experiments are conducted on DAVIS, UCF-101 and VIMEO90K-T.

**Strengths:**

1) VDP presents a simple and intuitive framework for accomplishing test-time, task-agnostic video editing.

2) The evaluation across multiple video tasks lends strong evidence to the generality of the proposed framework.



**Weaknesses:**

1) Lack of novelty: the propagation of video latent codes for editing tasks such as object removal, super-resolution, video interpolation etc. have been explored in previous works, such as [1][2]. The perceptual similarity and total variation losses have been borrowed from previous works without modification, and the spatial pyramid loss is simply a multi-level reconstruction loss, which has also been leveraged in previous works, such as [3][4][5][6].

2) Experimental setup and evaluation results are rather weak. For specific concerns, see the following questions.

[1] [Video-Specific Autoencoders for Exploring, Editing and Transmitting Videos](https://arxiv.org/abs/2103.17261v1)

[2] [Temporally Consistent Semantic Video Editing](https://arxiv.org/abs/2206.10590)

[3] [Loss Functions for Neural Networks for Image Processing](https://arxiv.org/abs/1511.08861)

[4] [Training a Task-Specific Image Reconstruction Loss](https://arxiv.org/abs/2103.14616)

[5] [Learning Multi-Scale Photo Exposure Correction](https://arxiv.org/abs/2003.11596)

[6] [Masked Image Modeling with Local Multi-Scale Reconstruction](https://arxiv.org/abs/2303.05251)

**Questions:**

1) Quantitative evaluation includes only PSNR and SSIM. However, these metrics have been identified in previous works to be problematic with respect to human-aligned notions of perceptual quality [1][2]. This problem is exacerbated for longer spatiotemporal sequences in video. Ideally, the authors could include user studies like [2][3]; in the absence of this evaluation, other metrics like FID, IS, LPIPS, etc. should be reported.

2) Fundamentally, I'm not convinced that the suite of tasks chosen by the authors is indeed the best for demonstrating the strength of their framework. Concretely, while the setting of explicitly removing dependency on external data is an interesting one, I'm wondering if this is a somewhat artificial constraint, particularly for tasks such as video super-resolution, interpolation, denoising, or object removal. Such tasks already have a well-established corpus of strong pretrained generative models which demonstrate convincing zero-shot performance on new images/videos. Further works similarly perform a few iterations of test-time optimization on single images/videos to improve performance on specific downstream tasks, as in [4][5]. Why should these video tasks be treated from a "external data"-free paradigm?

3) Questions 1) and 2) notwithstanding, the baselines used for comparison in each of the tasks are incomplete. For example, two strong prior works in [6] and [7] provide similar approaches to task-agnostic, semantic editing of videos, directly at test-time. These should be included for each of the tasks.

[1] [Complex Wavelet Structural Similarity: A New Image Similarity Index](https://pubmed.ncbi.nlm.nih.gov/19556195/)

[2] [The Unreasonable Effectiveness of Deep Features as a Perceptual Metric](https://arxiv.org/abs/1801.03924)

[3] [DreamSim: Learning New Dimensions of Human Visual Similarity using Synthetic Data](https://arxiv.org/abs/2306.09344)

[4] [Zero-Shot Video Editing Using Off-The-Shelf Image Diffusion Models](https://arxiv.org/abs/2303.17599v1)

[5] [MyStyle: A Personalized Generative Prior](https://arxiv.org/abs/2203.17272v1)

[6] [Blind Video Temporal Consistency via Deep Video Prior](https://arxiv.org/abs/2010.11838)

[7] [Learning Blind Video Temporal Consistency](https://arxiv.org/abs/1808.00449)



**Limitations:**

Please see the above questions with respect to limitations.

---

> ### Author Rebuttal · Authors · 2023-08-09
>
> We thank the reviewer for taking their time out from a busy schedule and evaluating our submission. We greatly appreciate that reviewer fnUo finds our framework simple and intuitive for performing video enhancement tasks. We are glad that the reviewer finds our framework generalizable to multiple video tasks. Below we address specific weakness and questions of the reviewer.
>
> **Weakness addressed:**
>
> We agree that previous works such as [3][4][5][6] have implemented a multiscale approach, however, their scope was mainly restricted to the spatial domain. In contrast, our contribution is the proposal of a novel spatial pyramid loss that extends this exploration of it’s properties in both the spatial and temporal domains. This combined utilization of spatial pyramid loss, perceptual loss, and TV loss is unique to our work and has allowed us to achieve results on par with, or superior to, state-of-the-art baselines, even those that are trained on external data. We believe this constitutes a significant leap forward, as it demonstrates that careful formulation of the loss function and model architecture can lead to solutions that match or exceed the performance of current data-driven methods, but without their inherent biases.
>
> Further highlighting our novelty, our approach demonstrates robustness against dataset biases. We showed this by introducing mild noise to the dataset and then conducting enhancement tasks, noting a significant drop in the performance of baselines trained using a dataset. This test underscores the strength of our method - the capability to maintain performance even when test data distribution varies slightly from the training data.
>
> Regarding work [2], we appreciate the mention, but we believe there is no overlap with our research.
>
> As for work [1], we weren't aware of this arXiv submission and will include it in our related work section. It indeed explores a video-specific approach, but there are key differences from our work. Firstly, for their reprojection property to function, they require a non-corrupted portion of the same video sequence, which may be less practical for enhancement tasks such as denoising and super-resolution. Secondly, their approach lacks a component to model temporal coherency beyond the proposed reprojection property. Thirdly, they rely solely on the L1 reconstruction loss to find a video-specific manifold, placing their approach in the L1 loss regime depicted in Fig. 2(iii) of our submission. This implies their solution's performance may significantly deteriorate in the presence of video signal noise. We appreciate your careful review and hope that these clarifications demonstrate the novelty and contributions of our work, warranting reconsideration of the reject rating.
>
> **Questions addressed:**
>
> **Metrics reporting:** We observed a general trend of reporting the PSNR and SSIM metrics in the previous papers; hence, we included only these metrics in our main paper. However, we also included other metrics like LPIPS and FVD metrics. These metrics are reported in the supplementary material in Table 5. It can be seen that our method performs better than current SOTA baselines.
>
> **Why an external data-free paradigm?** All the external corpus of the dataset is collected using some or other artificial constraints. For example, the Vimeo 90K dataset, which is commonly used to train video super-resolution baselines. The low-resolution frames in this dataset are devoid of noise or other artifacts arising after compression, a condition that seldom holds in real-world scenarios. Consequently, baselines trained on such datasets are optimally effective only under artifact-free conditions, limiting their utility in real-world settings.
>
> In contrast, our approach is not constrained by these artificial conditions, making it more robust against artifacts arising from non-ideal capture settings. We aim to demonstrate that a well-formulated model and loss function, independent of any training data, can deliver results on par with, if not superior to, the current data-driven methods.
>
> Although our method could certainly benefit from a strong text-to-image model or robust image baselines, we considered such enhancements to be beyond the scope of this submission.
>
> **Q3:** Please refer to the table in rebuttal pdf.

---

> > ### Comment · Reviewer_fnUo · 2023-08-15
> >
> > I thank the authors for their response. I appreciate the additional results, and pointing to the additional metrics in the supplementary material (it would be great if these could be incorporated into the main text in the revised/final draft.)
> >
> > However, I am still unconvinced of the fundamental setting underlying this approach. I still feel that the external data-free paradigm is a somewhat artificial constraint, particularly for the set of tasks portrayed in this work (super-resolution, interpolation, denoising, object removal). For example, I don't think it's true that super-resolution works trained on curated datasets are "limited in utility in real-world settings"; one only needs to take a perfunctory glance at the multitudes of GitHub repositories performing various forms of in-the-wild video enhancement with pretrained models to undermine this point.
> >
> > So I agree with the authors that incorporating more robust baselines is out of the scope of the submission, if only because I feel that such generative baselines _already_ offer concretely better performance on all the tasks depicted in this work. Thus, I maintain my original rating.

---

> > > ### Author Response · Authors · 2023-08-17
> > >
> > > Dear Reviewer,
> > >
> > >
> > > Thank you for your thoughtful feedback. Addressing your concerns in order:
> > >
> > >
> > > **External Data-Free Paradigm:**
> > > Firstly, while many methods perform impressively on curated datasets, our experience and extensive experiments suggest a general vulnerability of these models to slight changes in the data distribution. Yes, there exist numerous GitHub repositories that claim in-the-wild video enhancements (open-sourcing their pre-trained models). Still, these models often carry with them subtle biases from their training data. When faced with real-world scenarios deviating from their training context, they often falter and produce subpar results which we demonstrated through a series of experiments in our submission.
> > >
> > >
> > > Our central argument is not that traditional models fail but that their reliance on external training data makes them susceptible to unknown biases. Our proposed framework seeks to diminish such biases by adopting a training mechanism that's adaptive to the specific video at hand.
> > >
> > >
> > > **Integration of Robust Baselines:**
> > > Our reference to leveraging existing image-based baselines was conceptualized within the synergy of our approach. We did not imply that standalone image-based methodologies would outperform our model in video-specific tasks. While image baselines excel in single-frame tasks, their extrapolation to video sequences induces flickering artifacts. The prior works [6] and [7], as you rightly pointed out, attempt to integrate these image baselines by adding a temporal consistency factor. However, the results, when benchmarked against our approach, do not match up, as evidenced by the table in our rebuttal PDF.
> > >
> > >
> > > We hope this response elucidates our methodology's distinctiveness and addresses your concerns. We respect your viewpoint and deeply appreciate the time and effort you've invested in reviewing our work.
> > >
> > > Sincerely,
> > >
> > > Authors

---

> > > > ### Comment · Reviewer_fnUo · 2023-08-17
> > > >
> > > > I think the authors are not construing my criticism correctly. Concretely, I am asking for direct comparisons to strong pretrained generative models, not the weaker baselines employed in the various demonstration tasks (such as M2F2 or FastDVDNet used in the denoising task). For all the baselines included in the various video tasks within this work, am I correct in that none of them use state-of-the-art generative image/video models (i.e. GAN-based or diffusion-based approaches)? For example, how would the proposed approach compare against something like MCVD[1], with open source code + checkpoints?
> > > >
> > > > Yes, it's true that such pretrained models are not "external data" free, but my point is that a) the authors have not demonstrated through their experiments that such models are "susceptible to unknown biases," and b) it's not clear to me that even if such biases were present that they would induce inferior performance on standard video tasks like denoising, interpolation, etc. Thus, the particular motivation for _why_ we should consider "external data-free" a reasonable constraint is not well-supported by the experimental suite.
> > > >
> > > > [1] [MCVD: Masked Conditional Video Diffusion for Prediction, Generation, and Interpolation](https://arxiv.org/abs/2205.09853)

---

> > > > > ### Author Response · Authors · 2023-08-21
> > > > >
> > > > > Dear Reviewer,
> > > > >
> > > > > Thank you for the clarification. We understand your concerns better now, and we'll do our best to address them:
> > > > >
> > > > > **Comparison with baseline MCVD:** We attached the qualitative comparison(external link) with the baseline mcvd in the comment to AC. We urge AC to share this link with the reviewer. Notably, MCVD has an inclination to modify the original content within a scene. Such alterations significantly detract from the authenticity and fidelity of the resultant output. Moreover, the pretrained MCVD model is inherently constrained in its capability, specifically in handling resolutions beyond 64x64. These inherent shortcomings collectively contribute to its subpar performance, especially when compared with our framework. Such findings, derived from direct comparative analysis, serve to further highlight and emphasize the robustness of our proposed method.
> > > > >
> > > > > We genuinely value your engagement and the constructive insights you've shared.
> > > > >
> > > > > Sincerely,
> > > > >
> > > > > Authors

---

### Official Review · Reviewer_U9TZ · 2023-07-03

**Soundness:** 3 good
**Presentation:** 4 excellent
**Contribution:** 4 excellent
**Rating:** 7
**Confidence:** 4

**Summary:**

Authors of the paper propose a novel framework for low-level video enhancement tasks (i.e., denoising, object removal, frame interpolation, and super resolution).

The proposed framework does not require any training data by directly optimizing the neural model parameters over the corrupted test data.

The key idea is to design modules and loss terms that leverage the spatiotemporal coherence and internal statistics of the test videos - fundamental properties of a video sequence.

The proposed approach, the VDP model, consisting of a frame encoder and a frame decoder, performs future frame prediction.

The total loss to optimize this method at the time of use (test / inference time) is a combination of (1) the typical reconstruction loss, (2) a novel spatial pyramid loss, and (3) a variational loss introduced in a prior work.

VDP is more robust to spatio-temporal noises that contain in the input, and superior performance of the proposed approach was shown through qualitative results, quantitative results, comparisons to the state of the art (SOTA), and ablation studies on four low-level video enhancement tasks (i.e., denoising, object removal, frame interpolation, and super resolution).


**Strengths:**

S1 - Novelty:
A novel inference time optimization technique was proposed to learn neural module parameters / weights for performing various low-level video tasks.

S2 - Significance:
The proposed method eliminates the need for training data collection. It differs from the popular paradigm and is practical.

S3 - Quality:
Detailed analyses and experimental results have demonstrated the superiority of the proposed method.

S4 - Clarity:
The paper is well-constructed and easy to read.


**Weaknesses:**

W1 - The VDP model uses LSTM as the Latent Frame predictor Network and the choice is not well justified, although the authors have acknowledged this as a limitation of their current model and encourage readers to explore more architecture choices.

W2 - Oftentimes, when the input contains no noise, the proposed approach is on par with prior SOTA.

W3 - The resolution involved is still quite low (e.g., 3×448×256). Not sure if the proposed method can only handle resolution of this scale, or it can actually scale to high-resolution videos.

W4 - The test videos shown in the paper have relatively simple dynamics and a short time window. E.g., for frame interpolation and object removal. When the video has more complicated interactions and dynamics, will the model still perform well ? This is not clear.


**Questions:**

Q1: Is the proposed method the first inference time optimization technique for video tasks?

Q2: Is the optimization performed on the test set (rather than a single test video) ? If so, would the size of the test set be a factor greatly influencing performance of the VDP model ? What is the size of the VDP model in terms of the number of parameters ?

Q3: In section A of the Supplementary Material, line 30-31: ... for the frame interpolation task, the case is reversed, and cascaded baseline approaches yield better results than only baseline approaches.
Why would the denoiser wash away important details for the super-resolution task, but not frame interpolation task ?

Q4: What could be other possible low-level video tasks that might benefit from ideas of this paper ? How about other video tasks that focus on pixel-level spatio-temporal dynamics, e.g., tracking and particularly tracking any points (https://deepmind-tapir.github.io/) ? Are the proposed key components of the paper mostly just for video tasks that require spatio-temporal inpainting because of the future frame prediction based formulation ?

Q5: When the input contains no noise, what would be the reasons for someone to choose the proposed method over prior SOTA ?

Q6: In Table 4 of the Supplementary Material, since the inference time was given as the time to process one frame of a video, would it mean that the total time (Train + Infer) for the denoising task, for example, would be 3450 frame * 12 s / 3600 ~= 12 hours,  which is also the time for the VDP denoiser to be plateaued after 3600 epochs of training ?

Q7: Could you explain the following sentence ? Line 245: to perform a 4x frame interpolation, we select α = [0.25, 0.5, 0.75] in the Eqn. 5.


**Limitations:**

The auhtors have discussed the limitations. However, the proposed model might be also limited in its capability to handle videos with a high resolution, complex dynamics and a long time window.

---

> ### Author Rebuttal · Authors · 2023-08-09
>
> We thank the reviewer for taking their time out of a busy schedule and evaluating our submission. We greatly appreciate that reviewer U9TZ finds our approach novel, practical, and different from the popular paradigm. We are happy to note that the reviewer found the analysis of our approach detailed and our experimental evaluation comprehensive. We are glad that the reviewer understands the impact of our approach and how significant is the leap of getting SOTA results without the need for requiring external data. Lastly, we are glad the reviewer finds our paper well-constructed and easy to follow. Below we address specific weaknesses and questions of the reviewer.
>
> **Weakness Addressed:**
>
> **Why LSTM:**  Our choice of LSTM was motivated by its inherent ability to capture temporal dynamics in sequential data, making it a suitable choice for processing video data in our context. The memory cell in LSTM provides an efficient way to encode temporal information across frames, which is critical for video processing tasks such as denoising, super-resolution, and frame interpolation.
>
> **With no input noise the approach is at par with sota:** This is true with a caveat. Most SOTA baselines are trained and evaluated on similar datasets, which might not adequately reflect the diversity and unpredictability of real-world data. We have observed that the slightest introduction of noise into the input leads to a considerable drop in the performance of these baseline models, indicating their performance is closely tied to the specific characteristics of their training data. In contrast, our approach has demonstrated greater robustness against such perturbations, as it does not rely on any specific training data. This distinct advantage highlights the potential of our method to maintain high performance across diverse and realistic scenarios, providing more reliable and robust solutions.
> Is the approach limited by frame resolution: one simple way to scale the resolution is to add a bilinear interpolation layer before the final output layer. This way you can work with any resolution videos.
>
> **Handle complicated videos:** In our study, we utilized publicly available datasets widely acknowledged and used in previous literature. These datasets comprise videos with a diverse range of dynamics and interactions, thus offering a comprehensive testing environment for our model.
>
> **Questions:**
>
> **Are we the first internal learning approach for videos?** There exist some past internal learning approaches for videos that work on particular tasks, such as video object removal (referenced in our paper[23,53]) and traditional approaches (referenced in our paper[33,47]).
>
> **Would the size of the test set affect the performance?** The performance of our model is independent of the test set size. Number of parameters for our model is approximately 40 million.
>
> **Why the result changes with VSR and VFI when using cascaded models?** We think this might happen because the number of pixels to be predicted by the model increases 16 times in VSR(4x) task as opposed to 2 times in VFI(2x) task. So if the details are washed away by denoiser, this produces eight times more error in VSR than VFI.
>
> **Can this be extended to other approaches like point-tracking?** Our model is built on the idea of keeping the spatio-temporal consistency in a video which is also an essential component of point-tracking. So we do not see a reason why it would not be possible to extend this approach for such tasks. But at this point we considered this integration to be beyond the scope of the current paper.
>
> **Q5:** In a perfect world where no compression is taking place and all the videos are captured with ideal settings then it might be advantageous to use a trained baseline and achieve quick results.  However, this constraint does not hold true and artifacts are introduced because of compression of videos or non-ideal capture settings.
>
> **Q6:** Yes that calculation seems correct.
>
> **Q7:** When we say 4x frame interpolation, it means that for every single original frame, the system generates three additional frames to fit in-between the original ones, resulting in a video that plays at four times the original frame rate. These 3 frames are inserted at [0.25,0.5,0.75] intervals.

---

> > ### Comment · Reviewer_U9TZ · 2023-08-21
> >
> > I appreciate the authors' response. **It is not clear to me why the performance of the proposed method is independent of the test set size.**
> >
> > Optimization is performed on a test set, right? I assumed the optimization was carried out using Adam, in the way that is typically used in many traditional ML model optimizations. This usually involves iterating through the dataset multiple times, epoch by epoch, until the model converges. However, in your case, the optimization is done using the test set, with the assumption that there's no training set.
> >
> > If the calculation in Q6 is correct, then a larger test set would allow the model's weights to be optimized using more test examples. Intuitively, this could lead to longer optimization time and potentially improved performance. I'm confused about the optimization process after reading the response.
> >
> > Some questions are not answered:
> >
> > **What could be other possible low-level video tasks that might benefit from ideas of this paper?** This question shares the concerns of reviewer fnUo, i.e., whether the proposed setup or method is just particularly for tasks demonstrated such as video super-resolution, interpolation, denoising, or object removal.
> >
> > **Authors mentioned the proposed method can be applied to point-tracking, but how? Could authors elaborate?** This would further ease the concern on the scope of the proposed method (or setup).

---

> > > ### Author Response · Authors · 2023-08-21
> > >
> > > Dear Reviewer,
> > >
> > > We deeply appreciate your meticulous review of our submission and the constructive feedback provided. We provide further clarification on the issues raised.
> > >
> > > **Test Set Size and Performance:**
> > > Your understanding of our method is right about not requiring the training dataset and the calculations raised in Q6. However, for our method, the optimization process is executed independently for each video, meaning that we optimize for a specific video until convergence. Once completed, we obtain the enhanced video, and the process restarts for the next video. Thus, our model's nature involves treating each video in the test set as an isolated case. Hence,  the collective test set size does not influence the performance of the model for any specific video.
> > >
> > >
> > > **Application to Other Low-Level Video Tasks:**
> > > Our framework's intrinsic nature emphasizes strict spatio-temporal consistency, as portrayed in Figure 2(iii). For pixel correspondence tasks like point-tracking, an intuitive extension could involve a synergistic approach, combining our technique with a DINO[1][2] penalty on the video. Here, the DINO penalty can assist in determining inter-frame pixel correspondence, while our method ensures enhanced spatio-temporal coherence, potentially leading to more accurate tracking results. However, delving into this research direction is expansive and beyond the current paper's scope, even though it signifies an exciting avenue for future exploration.
> > >
> > >
> > > Hopefully, our response clarifies all the questions raised. We thank you once again for your constructive feedback. We're committed to refining our submission further, taking into account the suggestions provided.
> > >
> > > Sincerely,
> > >
> > > Authors
> > >
> > >
> > > [1] Emerging Properties in Self-Supervised Vision Transformers
> > >
> > > [2] DINOv2: Learning Robust Visual Features without Supervision

---

> > > > ### Comment · Reviewer_U9TZ · 2023-08-21
> > > >
> > > > Thanks for the response!

---

### Official Review · Reviewer_Pc2m · 2023-07-05

**Soundness:** 3 good
**Presentation:** 2 fair
**Contribution:** 3 good
**Rating:** 5
**Confidence:** 4

**Summary:**

This work proposed an internal learning method for multiple video-processing tasks. Without collecting large-scale training data, the proposed method can achieve comparable or even better results than baselines. But the slow computation is one obvious limitation.

**Strengths:**

- The proposed method is optimized individually for each video and thus does not need to collect training data.
- Extensive experiments on different tasks such as super-resolution, denoising, and inpainting show effectiveness.

**Weaknesses:**

- As the proposed method is optimized for each video, I believe that it will work on many examples and it will also fail on many examples. How about the generalization ability of the proposed method? Are results in the paper cherry-picked or randomly sampled? How about failure cases?
- The writing for the technical part can be further improved. The current version just  lists some equations and the procedures, and more motivation, explanation, insights would be better for readers.
- One common problem of internal learning methods is that it would fail if there is insufficient information within a video. For example, if some objects keep blurry across the videos, it is difficult to recover it with internal learning. How could you solve this problem? Could the proposed method be combined with some pretrained models?
- The optimization process is slow

**Questions:**

see weakness

**Limitations:**

see weakness

---

> ### Author Rebuttal · Authors · 2023-08-09
>
> We thank the reviewer for taking their time out of a busy schedule and evaluating our submission. We greatly appreciate that reviewer Pc2m finds the effectiveness of our approach while requiring no external data as one of the strengths of our approach. We are happy to note that the reviewer found our experimental section comprehensive. Below we address specific weaknesses and questions of the reviewer.
>
> **Weakness addressed:**
>
> **Failure cases** primarily emerge when crucial information about the object of interest in a video is not present or discernable. For instance, we draw your attention to the provided figure in the attached rebuttal PDF. This figure demonstrates a scenario where we first downscaled the original sequence by 4x, thereby eliminating a significant amount of discernable facial information across the frames. Consequently, in the subsequent super-resolution process, our model was unable to recover all lost details, as visualized in the same figure. However, this is not a unique limitation to our approach but is also observed with other baseline methods, which is evident from the figure.
> For the demo videos for our method, we assure you that they were not cherry-picked. We also provide quantitative results in our submission to back the evaluation of our approach. These quantitative evaluations take into account both the success and failure cases in our model performance.
> **Pretrain model usage:** In cases where a pretrained image model exists, its output could potentially be incorporated with our model to produce spatiotemporally consistent results. While we acknowledge this potential avenue, we consider this integration to be beyond the scope of the current paper.
>
> **Better motivation:** We will work on a more nuanced motivation in our introduction section of the paper by explaining the downside of relying on a data-driven pipeline and how our approach tries to resolve such issues by removing dataset biases. Building on suggestions from reviewer 6LSS, we would also incorporate how changing the architecture for FDNet and LFPNet affect the denoising regime(represented by the Fig 2(iii)) of our VDP model.
>
> **Slow optimization process:** Our primary goal in this submission was to establish the efficacy and robustness of our approach without using any external data, and we believe we have achieved that. While we acknowledge slow optimization as a current limitation, we view it as an important area for future improvements. Future work could explore more efficient optimization techniques or parallel computation strategies to accelerate the process.

---

> ### Comment · Reviewer_Pc2m · 2023-08-18
>
> Thanks to the authors for their responses. After reading the rebuttal and comments from other reviewers, I would keep my original ratings. I would suggest authors including a visualization of different failure cases in their revised version.

---

> > ### Author Response · Authors · 2023-08-21
> >
> > Dear Reviewer,
> >
> > We sincerely appreciate you taking the time to review our submission and for considering the points made in our rebuttal. Your positive feedback is invaluable, and we are committed to incorporating your suggestions in the revised version of our paper.
> >
> > Sincerely,
> >
> > Authors

---

### Official Review · Reviewer_6LSS · 2023-07-07

**Soundness:** 3 good
**Presentation:** 4 excellent
**Contribution:** 3 good
**Rating:** 7
**Confidence:** 4

**Summary:**

This paper introduces a new inference time optimization approach for low-level image enhancement tasks. The proposed approach does not need any training data and leverages the spatio-temporal coherence and internal statistics of test videos. The authors also propose a spatial pyramid loss which makes the proposed approach robust to spatial and temporal noise. The proposed approach is evaluated on multiple video enhancement tasks (denoting, frame interpolation, super-resolution, and object removal) and is shown to perform competitively or better than existing works that use a training dataset.

**Strengths:**

The paper is written well and was easy to follow
The proposed approach is applicable to many video enhancement tasks without the need for training task-specific networks on large training datasets
Experimental evaluation on several datasets clearly show the effectiveness of the proposed approach.

**Weaknesses:**

Since the proposed approach trains a network on the test video, the results depend on how long it is trained and the required training steps may vary from video to video depending on the content. If we train long enough, the model may start fitting to the corruptions in the input video to some extent (though the losses are providing impedance). It is unclear to me if the results presented in the paper correspond to using the same number of steps for all videos, or if the authors determined when to stop based on the loss curve for each video independently.

The performance of the proposed approach also depends on the network architecture used. Authors haven't conducted experiments to show how sensitive the results are to the choice of architectures of LFPNet and FDNet. Different architectures may have different inductive biases and may have an impact of the behavior of loss observed in Figure 2(iii).

**Questions:**

See my response in the weaknesses section.

**Limitations:**

The main limitation of the proposed approach is that it is an offline approach. Authors have clearly acknowledged the limitations of their approach and discussed potential ways to address them (in the supplementary material).

---

> ### Author Rebuttal · Authors · 2023-08-09
>
> We thank the reviewer for taking their time out of a busy schedule and evaluating our submission. We are glad reviewer 6LSS found our paper well-written and easy to follow. We greatly appreciate that reviewer 6LSS finds our approach versatile on many video tasks while requiring no external data. We are happy to note that the reviewer found our experimental section comprehensive. Below we address specific weaknesses and questions of the reviewer.
>
> **Weaknesses addressed:**
>
> To ensure a fair and consistent comparison across different videos, we maintained a uniform number of training steps for all the videos in our test dataset. We mentioned the number of epochs required per task in our supplementary material Sec. D. Specifically, the optimization was executed for the mentioned number of epochs for each video, irrespective of the video content. Our presented results align with this uniform number of training steps across all videos, guaranteeing consistency in our approach. The choice of these epochs were made such because the performance of VDP plateaued after these many epochs. Thus, the potential concern raised by the reviewer about varying the training steps depending on the loss curve for each video independently was not an issue in our experimentation due to our consistent approach.
>
> **Denoising regime could be different for different architectures:** We thank the reviewer for raising this important point as exploring better architecture for FDNet and LFPNet in itself can be a good research problem. We explored two settings of FDNet (one with residual connections and another without them just the vanilla conv decoder) and LFPNet ( one with lstm cells and the other with vanilla RNN cells). The denoised regime of these can be seen from the plot in the rebuttal pdf. It can be observed from the plot that vanilla conv decoder based FDNet and rnn based LFPNet have a suboptimal denoised regime than our current design for FDNet and LFPNet.

---

> > ### Comment · Reviewer_6LSS · 2023-08-14
> > **Thanks you for the rebuttal.**
> >
> > After reading the rebuttal, I stick to my initial rating which is in favor of accepting the paper.

---

> > > ### Author Response · Authors · 2023-08-17
> > >
> > > Dear Reviewer,
> > >
> > > Thank you for taking the time to review our submission and going through our rebuttal. We greatly appreciate your positive feedback and will incorporate your suggestions in the revised version of our submission.
> > >
> > > Sincerely,
> > >
> > > Authors

---

### Author Rebuttal · Authors · 2023-08-09

Extra results in pdf file.

---

### Decision · Program_Chairs · 2023-09-21

**Decision:**

Accept (poster)

**Comment:**

After the review and rebuttal three reviewers argue for acceptance, one for rejection. All reviewers agree that the current experimental validation is limited, but three reviewers appreciate the technical novelty, while one reviewer found the presented work to be too close to prior work. After reading the paper and flagged related work, the AC agrees with three reviewers that the presented work as sufficiently different to warrant acceptance.